# Inflammatory Biomarker Score Identifies Patients with Six-Fold Increased Risk of One-Year Mortality after Pancreatic Cancer

**DOI:** 10.3390/cancers13184599

**Published:** 2021-09-13

**Authors:** Alisa D. Kjaergaard, Inna M. Chen, Astrid Z. Johansen, Børge G. Nordestgaard, Stig E. Bojesen, Julia S. Johansen

**Affiliations:** 1Steno Diabetes Center Aarhus, Aarhus University Hospital, 8200 Aarhus, Denmark; 2Department of Oncology, Herlev and Gentofte Hospital, Copenhagen University Hospital, 2730 Herlev, Denmark; inna.chen@regionh.dk (I.M.C.); astrid.zedlitz.johansen@regionh.dk (A.Z.J.); julia.sidenius.johansen@regionh.dk (J.S.J.); 3Department of Clinical Biochemistry, Herlev and Gentofte Hospital, Copenhagen University Hospital, 2730 Herlev, Denmark; boerge.nordestgaard@regionh.dk (B.G.N.); stig.egil.bojesen@regionh.dk (S.E.B.); 4Department of Clinical Medicine, Faculty of Health and Medical Sciences, University of Copenhagen, 2200 Copenhagen, Denmark

**Keywords:** C-reactive protein, CA-19-9 antigen, interleukin-6, chitinase-3-like protein 1, carcinoma, pancreatic ductal

## Abstract

**Simple Summary:**

For 20 years, the CA 19-9 blood test has been the only broadly used biomarker of pancreatic ductal adenocarcinoma (PDAC). We lack easily available biomarkers to help differentiate patients between good, intermediate and poor survivors at the time of PDAC diagnosis. Using one of the largest studies of patients with PDAC, we found that a simple combination of blood tests, namely CRP, CA 19-9 and IL-6, into a single biomarker score was a better marker of one-year survival than the currently recommended CA 19-9 alone or any other combination of the four inflammatory biomarkers examined (CRP, CA 19-9, IL-6 and YKL-40). However, since this is the first study examining this inflammatory biomarker score, future validation studies are needed. Moreover, CRP outperformed CA 19-9 in the majority of patients, thus questioning the routine use of CA 19-9 in patients with PDAC.

**Abstract:**

We examined whether elevated plasma C-reactive protein (CRP), carbohydrate antigen (CA) 19-9, interleukin-6 (IL-6) and YKL-40, individually or combined, can identify poor survivors among patients with pancreatic ductal adenocarcinoma (PDAC). We measured CRP, CA 19-9, IL-6 and YKL-40 in 993 patients at the time of PDAC diagnosis. The biomarker score was the sum of biomarker categories, coded 0, 1 and 2 for low, intermediate and high plasma concentrations, respectively. High vs. low levels of CRP, CA 19-9 and IL-6 were each independently associated with a two-fold increased risk of one-year mortality. CRP performed best in patients with advanced and CA 19-9 in patients with low cancer stages. YKL-40 was not associated with mortality and, therefore, was not included in the biomarker score. Compared to the biomarker score = 0, the multifactorially adjusted hazard ratios for one-year mortality were 1.56 (95% confidence interval: 0.99–2.44) for score = 1, 2.22 (1.41–3.49) for score = 2, 3.44 (2.20–5.38) for score = 3, 5.13 (3.21–8.17) for score = 4 and 6.32 (3.84–10.41) for score = 5–6 (*p*-value for trend = 3 × 10^−31^). This score performed better than any single biomarker or combination of biomarkers when examined in similarly sized or other categories. In conclusion, a combination score of elevated CRP, CA 19-9 and IL-6 identified patients with six-fold higher one-year mortality.

## 1. Introduction

Pancreatic cancer is currently the fourth and is projected to become the second leading cause of cancer deaths in both the United States of America and Europe (https://gco.iarc.fr, accessed on 2 August 2021) [1,2]. Global incidence and mortality rates are slightly higher in men than in women, and historical incidence trends coincide with rising prevalences of smoking, overweight and diabetes [1,3]. There are usually no or few symptoms at early stages, and pancreatic cancer is therefore often diagnosed at a late stage [4]. Hence, the chances for complete resection and curative treatment are slim, and the overall five-year survival is only 9%, with little improvements over the last years [1]. Importantly, we lack easily available biomarkers to help differentiate patients between good, intermediate and poor survivors at the time of diagnosis of pancreatic cancer.

Inflammation plays an important role in the development and progression of pancreatic cancer [5]. Numerous previous studies reported that inflammatory biomarkers, including C-reactive protein (CRP) and interleukin-6 (IL-6), were independently associated with overall survival in (from only 7 to 474) patients with pancreatic cancer [6,7,8,9,10,11]. Previous results from the Danish BIOPAC (BIOmarkers in patients with PAncreatic Cancer) study on 559–592 patients with pancreatic cancer found that plasma CRP, carbohydrate antigen 19-9 (CA 19-9), IL-6 and YKL-40 (see Appendix A) may each be independent biomarkers of overall survival [12,13]. However, for more than 20 years, CA 19-9 has remained the only broadly used biomarker of pancreatic ductal adenocarcinoma (PDAC) [14]. A recent Finnish study of 212 patients who were operated on for PDAC showed that a combination of elevated CRP (≥3 mg/L) and CA 19-9 (>37 kU/L) was associated with a three- to four-fold increased risk of mortality [15]. Furthermore, this combination of elevated CRP and CA 19-9 outperformed combinations of elevated CRP and decreased albumin (original and modified Glasgow prognostic scores) as well as other routine biochemical analyses (leukocytes, platelets, bilirubin and carcinoembryonic antigen) for prognostic purposes [15].

We therefore tested the hypothesis that elevated plasma CRP, CA 19-9, IL-6 and YKL-40 individually and as a combined biomarker score are associated with poor survival in patients with PDAC.

## 2. Materials and Methods

This study adheres to the REMARK (Reporting Recommendations for Tumor Marker Prognostic Studies) guidelines.

We included 993 adult patients with histologically verified PDAC from the Danish BIOPAC (“BIOmarkers in patients with PAncreatic Cancer (BIOPAC)—can they provide new information of the disease and improve diagnosis and prognosis of the patients”; ClinicalTrials.gov ID: NCT03311776) study [16]. The BIOPAC study is a prospective multicenter open cohort study with ongoing enrollment. The patients in this study were enrolled at five hospitals: Copenhagen University Hospital at Herlev (50%, houses the biobank), Hillerød Hospital (6%), Zealand University Hospital at Næstved (9%), Odense University Hospital (10%) and Copenhagen University Hospital at Rigshospitalet (25%). Because Rigshospitalet is the most highly specialized hospital in Denmark and the largest one in volume with expertise in pancreatic cancer surgery, 71% of patients enrolled at Rigshospitalet were operated on. As operated patients usually have a smaller tumor burden and a better prognosis than patients who are not operated on, this likely explains why patients enrolled at Rigshospitalet had somewhat lower plasma levels of inflammatory biomarkers. However, no other patient characteristics differed between the hospitals. The enrollment period was from 3 July 2008 to 24 August 2017. The patients were followed until 4 October 2020, or death, whichever came first, for at least three years. Information on death was obtained from the national Danish Civil Registration System. The patients were operated on and/or treated with different types of chemotherapy according to national guidelines (www.gicancer.dk, accessed on 2 August 2021).

### 2.1. Biochemical Analyses

The blood for biochemical analyses was drawn around the time of diagnosis. For patients with stage II–IV PDAC, blood was drawn just prior to the commencement of chemotherapy (median (25th–75th percentile): 0 (0–0) days). For patients who were operated on (≈30%), it was drawn prior to the operation (0 (0–34) days). The biochemical analyses of CRP (*n* = 975), CA 19-9 (*n* = 977), IL-6 (*n* = 993) and YKL-40 (*n* = 993) were performed on the same serum sample per patient. The levels of CRP were measured by turbidimetry (Dako, Glostrup, Denmark) using a standard laboratory assay, subjected to internal (daily) and external (monthly) quality control programs. CA 19-9 was analyzed in serum samples using a solid phase, two-site sequential chemiluminescent immunometric assay, the Immulite 2000 GI-MA (catalogue Number L2KG12, Siemens, Ballerup, Denmark). Due to some extremely high CA 19-9 measurements (up to 1,665,063 kU/L), we set an upper limit for CA 19-9 quantification levels to 100,000 kU/L. IL-6 and YKL-40 were determined in serum samples in duplicates by commercial two-site, sandwich-type enzyme-linked immunosorbent assays (IL-6: Catalogue number HS600, R&D Systems, Abingdon, Oxon, UK; YKL-40: Quidel Corporation, San Diego, CA, USA). The detection limit was 0.01 ng/L for IL-6 and 20 µg/L for YKL-40.

### 2.2. Covariates

Information on baseline covariates was collected from patient records at the time of enrollment, i.e., blood sampling. We used participant-reported information on smoking habits and alcohol consumption. High alcohol consumption was defined as alcohol intake above 7 and 14 drinks per week for women and men, respectively (1 drink ≈ 12 g of alcohol). Body mass index was calculated by measured weight in kilograms (at enrollment) divided by measured height in meters squared. We calculated Charlson’s age adjusted comorbidity index as the Charlson’s comorbidity index after adding one point for each 10-year increase from 40 years of age [17]. Other covariates were performance status (PS), diabetes, operation, the presence of metastasis, tumor size and PDAC stage.

### 2.3. Statistical Analysis

We stratified the biomarkers into low, intermediate and high categories (percentage of the patients): CRP into <10 mg/L (51%), 10–100 mg/L (40%) and ≥100 mg/L (9%); CA 19-9 into <37 kU/L (19%), 37–10,000 kU/L (67%) and ≥10,000 kU/L (14%); IL-6 into <5 ng/L (47%), 5–25 ng/L (43%) and ≥25 ng/L (10%); and YKL-40 into <200 µg/L (68%), 200–799 µg/L (27%) and ≥800 µg/L (5%).

For CRP, CA 19-9 and IL-6, we chose commonly used routine cut-off levels, levels previously reported in the literature [7,18] and/or rounded even levels where approximately 10% of patients were in the high category. For YKL-40, the low and high cut-offs at <200 µg/L and ≥800 µg/L corresponded with previously used levels [19].

Interaction (between biomarkers and PS, operation and PDAC stage) was examined by the likelihood ratio test after linear regression between the models with and without an interaction term.

For each patient, we calculated a biomarker score_4_ as the sum of each of the four biomarkers, coded 0, 1 and 2 for low, intermediate and high biomarker categories, respectively. Thus, patients received a biomarker score_4_ between 0 (if all four biomarkers were low) and 8 (if all four biomarkers were high). Additionally, we excluded plasma YKL-40 from the biomarker score_4_ (as plasma YKL-40 was not associated with survival when adjusted for the three other biomarkers) and generated a new biomarker score_3_ of three biomarkers: CRP, CA 19-9 and IL-6 (score between 0 and 6).

Moreover, we used logarithmic base two transformations of CRP, CA 19-9, IL-6 and YKL-40 levels to estimate survival for a two-fold increase, i.e., doubling.

Because patients with PDAC have a very low five-year survival, we investigated cumulative survival for up to five years and the risk of one-year mortality. We used the Kaplan–Meier survival function to plot five-year cumulative survival curves, overall and stratified by PS (PS = 0 versus PS ≥ 1), operation (yes/no) and PDAC stage (stages I–II versus III–IV). The differences across biomarker categories and other categories of covariates were examined using log-rank trend tests. We used Cox regression analysis (with time since blood sampling as the time scale) to calculate hazard ratios (HRs) and 95% confidence intervals (CIs) for one-year mortality after PDAC.

Model 1 was adjusted for age and sex and Models 2 and 3 additionally for PS (PS ≤ 1 or PS ≥ 2), operation (yes/no), cancer stage (I–II, III or IV) and the three biomarkers other than the one examined. Model 2 included only patients with complete information on these covariates, while Model 3 included all patients after multiple imputation of missing covariates. We had information on age, sex, operation and plasma IL-6 and YKL-40 measurements on all participants. We imputed missing values (Table 1) of CRP (*n* = 18) by negative binomial regression; values of CA 19-9 (*n* = 16) by linear regression; and values of cancer stage (*n* = 11), PS (*n* = 96) and biomarker category (*n* = 18 for CRP and *n* = 16 for CA 19-9) by ordered logistic regression models using substantive model compatible fully conditional specification [20].

The predictive values of plasma biomarkers for one-year mortality were examined by area under the receiver operating characteristic curves (AUC).

### 2.4. Additional Analyses

We investigated pairwise combinations of CRP, CA 19-9, IL-6 and YKL-40. Furthermore, in order to investigate the recently proposed prognostic score [15], we grouped CRP and CA 19-9 levels into (1) CRP < 3 mg/L and CA 19-9 ≤ 37 kU/L, (2) CRP ≥ 3 mg/L or CA 19-9 > 37 kU/L and (3) CRP ≥ 3 mg/L and CA 19-9 > 37 kU/L. Finally, additional analyses regarding measured as well as genetically predicted plasma YKL-40 (*CHI3L1* rs4950928 genotype) in relation to CRP, CA-19-9 and IL-6 are described in Appendix A [19,21,22,23,24,25,26,27,28,29,30,31,32].

## 3. Results

Baseline characteristics are shown in Table 1. Thirty percent of the patients were operated on, as their tumor was considered resectable. During the study period, 947 patients (95%) died. The median follow-up time was 264 days (25–75th percentile: 129–557 days). The five-year survival was only 6.3% (63 out of 993 participants).

### 3.1. Plasma CRP, CA 19-9, IL-6, YKL-40 and Survival

For the increasing categories of all four biomarkers, overall survival after PDAC decreased (Figure 1 and Appendix A).

Poorer PS (PS ≥ 1 versus PS = 0), not being operated (versus being operated) on and a higher PDAC stage (stage III–IV versus stage I–II) were associated with decreased survival (Appendix A). We observed no robust interactions between the four biomarkers and PS, operation and PDAC stage. HRs (Model 3) for one-year mortality were 2.20 (95% confidence interval: 1.62–2.99) for high versus low CRP, 2.13 (1.56–2.92) for high versus low CA 19-9, 2.08 (1.51–2.89) for high versus low IL-6 and 1.07 (0.74–1.54) for high versus low plasma YKL-40 (Figure 2).

The HRs (Model 3) for one-year mortality per doubling in plasma biomarker levels were 1.18 (1.13–1.24) for CRP, 1.07 (1.05–1.10) for CA 19-9, 1.23 (1.15–1.31) for IL-6 and 1.06 (0.98–1.14) for YKL-40 (Appendix A).

In accordance with the observation that elevated plasma YKL-40 was not independently associated with poor survival after adjustment for the three other biomarkers, genetically elevated YKL-40 (chitinase-3-like-1 (CHI3L1) rs4950928 genotype) was not associated with poor survival either (Appendix A).

CRP, CA 19-9 and IL-6 were comparable with regard to the prediction of one-year mortality using the area under the receiver operating characteristic curves (AUROC), while the corresponding values for plasma YKL-40 were consistently smaller than those for any other biomarker (Figure 3).

CRP (and IL-6) had the best one-year survival predictive ability for patients who were not operated on and had stage III–IV PDAC, while CA 19-9 had the best one-year survival predictive ability for patients who were operated on and had stage I–II PDAC (Figure 3). When we restricted the analyses to participants who had measurements of all four biomarkers, the results were similar (Appendix A).

### 3.2. Plasma CRP, CA 19-9, IL-6 and YKL-40: Biomarker Score

Increasing categories of biomarker score were associated with increased mortality (Figure 4 and Appendix A).

Compared to the biomarker score_4_ = 0, the HRs (Model 3) for one-year mortality were 1.44 (0.90–2.28) for score_4_ = 1, 1.67 (1.05–2.66) for score_4_ = 2, 2.81 (1.77–4.49) for score_4_ = 3, 3.48 (2.18–5.56) for score_4_ = 4, 4.62 (2.80–7.62) for score_4_ = 5 and 5.49 (3.30–9.13) for score_4_ = 6–8; *p*-trend = 7 × 10^−28^ (Appendix A), corresponding to an HR of 1.34 (1.27–1.41) for each unit increase in biomarker score_4_.

When we excluded YKL-40 from the biomarker score_4_, i.e., restricted the score_3_ to CRP, CA 19-9 and IL-6, the estimates were augmented. Compared to the biomarker score_3_ = 0, the HRs for one-year mortality were 1.56 (0.99–2.44) for score_3_ = 1, 2.22 (1.41–3.49) for score_3_ = 2, 3.44 (2.20–5.38) for score_3_ = 3, 5.13 (3.21–8.17) for score_3_ = 4 and 6.32 (3.84–10.41) for score_3_ = 5-6; *p*-value for trend = 3 × 10^−31^ (Figure 5), corresponding to an HR of 1.46 (1.37–1.55) for each unit increase in biomarker score_3_.

When we stratified the analyses by operation status, the risk estimates for one-year mortality were even higher nominally (but with very broad 95% CIs) for the patients who were operated on compared to the patients who were not operated on (Appendix A). However, we found no evidence of interactions between biomarker score_3_ and PS, operation and PDAC stage (all *p*-values for interaction ≥ 0.09).

Importantly, the inflammatory biomarker score_3_ performed better than any of the four single biomarkers when examined in similarly sized or other categories.

### 3.3. Additional Analyses

The HRs (Model 3) for one-year mortality were similar for CRP in combination with either CA 19-9 or IL-6 (4.64 (2.60–8.30) and 4.69 (3.29–6.68) for score = 4 versus score = 0, respectively, Appendix A). The corresponding estimates were somewhat lower for pairwise biomarker combinations with YKL-40 and somewhat higher for a combination of CA 19-9 and IL-6 but with largely overlapping confidence intervals, and, thus, they were not robustly different from each other (Appendix A). Estimates were higher but imprecise for patients who were operated on compared to those who were not operated on (Appendix A).

Compared to CRP < 3 mg/L and CA 19-9 ≤ 37 kU/L, the HRs (Model 3) for one-year mortality were 1.51 (0.75–3.00) for CRP ≥ 3 mg/L or CA 19-9 > 37 kU/L and 2.62 (1.34–5.13) for CRP ≥3 mg/L and CA 19-9 > 37 kU/L (Appendix A). Estimates were higher but imprecise for patients who were operated on compared to those who were not operated on (Appendix A).

The results regarding plasma YKL-40 and *CHI3L1* rs4950928 in relation to CRP, CA-19-9 and IL-6 are shown in Appendix A).

## 4. Discussion

By studying 993 patients with PDAC from the BIOPAC study, we found that elevated plasma CRP, CA 19-9 and IL-6, but not YKL-40, were independently associated with poor survival. A combination score of elevated CRP, CA 19-9 and IL-6 identified patients with PDAC who had more than six-fold higher one-year mortality. Importantly, the biomarker score performed substantially better than each biomarker individually and pairwise. The ability of elevated CRP, CA 19-9 and IL-6 to predict one-year mortality, in terms of ROC statistics, was comparable to the ability of low-density lipoprotein cholesterol and apolipoprotein B to predict myocardial infarction [33].

Compared with previous studies based on the BIOPAC [12,13], the present study included twice as many patients and had a longer follow-up time. Because 95% of the participants died during follow-up, we assessed one-year survival, which we had complete information on. Furthermore, we stratified cumulative survival and AUROC analyses by PS, operation and PDAC stage. Finally, while the previous studies dichotomized biomarkers, we grouped biomarker levels into three categories each.

Our results are largely in agreement with the previous studies because (1) YKL-40 was not robustly associated with overall survival; (2) CA 19-9 was superior to IL-6 as a prognostic marker for patients who were operated on, while the opposite was the case for patients who were not operated on; and (3) a combination of biomarkers was a better prognostic marker than each biomarker individually [12,13]. Importantly, a combination of CRP, IL-6 and CA 19-9, and thus without YKL-40, had the best prognostic performance with regard to one-year mortality, particularly for the patients who were operated on.

In the present study, CRP and IL-6 performed best as prognostic biomarkers for patients who were not operated on and had high stage PDAC, while CA 19-9 performed best as a prognostic biomarker for patients who were operated on and had low stage PDAC. A possible explanation could be that CRP and IL-6 compared with CA 19-9 may be more sensitive biomarkers of other inflammation than PDAC, e.g., infections. However, this is only speculative. Nonetheless, our findings suggest that for the 70% of patients with PDAC who are not operated on, there is no reason to favour CA 19-9 above CRP for prognostic purposes. This is in agreement with previous studies that showed similar performance of the two biomarkers [7,11,18,34,35]. Measuring CRP is inexpensive and easily available and could be particularly useful for the 5–10% of patients who do not produce plasma CA 19-9 [36]. However, plasma CA 19-9 has remained the most widely used biomarker for PDAC [14] and the only biomarker currently recommended for clinical use by the US National Comprehensive Cancer Network guidelines (https://www.nccn.org/professionals/physician_gls/pdf/pancreatic.pdf, accessed on 2 August 2021).

Our results are largely in agreement with a recent Finnish study of 212 patients who were operated on for PDAC [15]. We found that elevated CRP (≥3 mg/L) and/or CA 19-9 (>37 kU/L) was associated with a two- to five-fold increased risk of one-year mortality in patients who were operated on, but there was no robust association in those who were not.

The strengths of this study include its prospective design, the large number of patients with PDAC and the measurements of four biomarkers of each patient. It is also a strength that we had 100% follow-up; that is, we did not lose track of even a single patient.

The potential limitations of the present study include survival bias and selective nonresponse by individuals with poor PS and a high PDAC stage. However, the lack of patients with the most severe phenotype would likely bias the estimates toward the null, and the five-year survival of only 6.4% does not support this. Another limitation is that we were unable to perform a validation study of the biomarker score. Despite the relatively large size of this study, when we stratified patients according to operation status, the confidence intervals for biomarker score were very broad, particularly in the operated strata. According to the Danish national guidelines, a conventional CT scan is the most commonly used modality at the initial staging of PDAC. Thus, microscopic disease and small metastases under the detection level may have been overlooked. Since very few patients had an additional PET-CT performed, we were unable to investigate CA 19-9 levels according to CT modality (conventional versus PET-CT). Furthermore, not only PDAC but also many infections are associated with increased plasma levels of inflammatory biomarkers. However, since blood sampling was performed just prior to the commencement of chemotherapy or operation, both of which are contra-indicated in patients with overt infections, it is unlikely that infections affected plasma biomarker levels to a large degree. Finally, as we only included patients from the (predominantly white) Danish population, our findings do not necessarily apply to other ethnicities. Yet, to our knowledge, there are no data suggesting that our findings could not also be directly applicable to other ethnicities.

## 5. Conclusions

In this large prospective study of patients with PDAC, we found that CRP out-performed CA 19-9 for prognostic purposes in the 70% of patients with PDAC who were not operated on. A combination of CRP, CA 19-9 and IL-6 into a single biomarker score_3_ was the best prognostic marker and may be clinically valuable in identifying patients with PDAC who have the poorest prognosis.

## Figures and Tables

**Figure 1 cancers-13-04599-f001:**
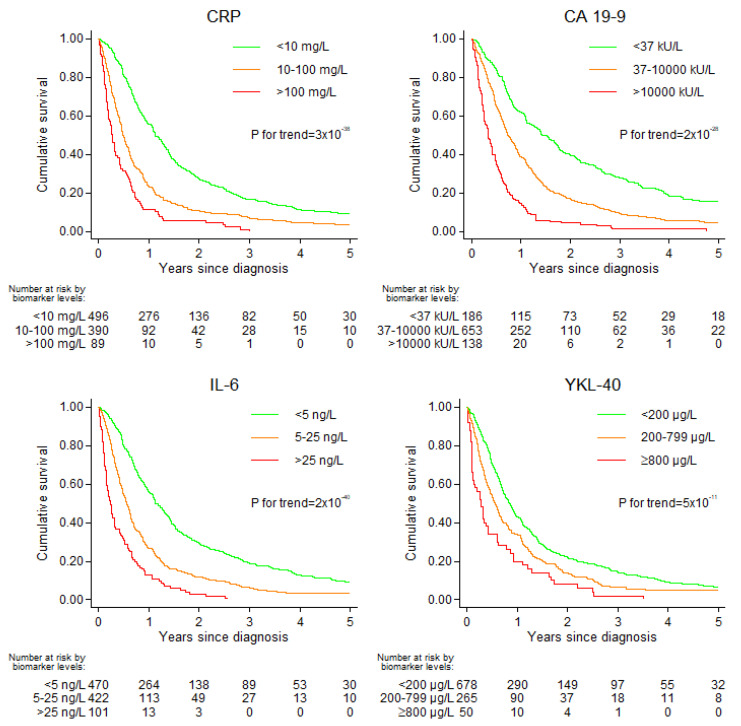
Kaplan–Meier survival curves for plasma CRP, CA 19-9, IL-6 and YKL-40 categories. *p*-value for trend is from Wald test of trend across groups.

**Figure 2 cancers-13-04599-f002:**
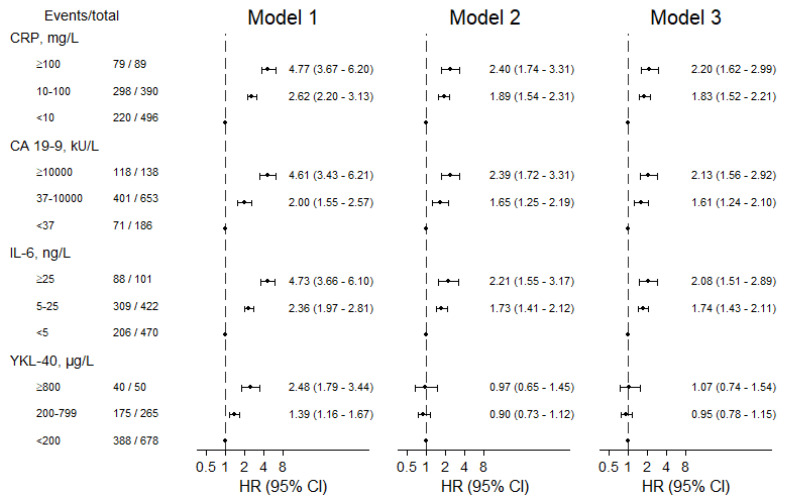
Risk of one-year mortality after diagnosis of pancreatic ductal adenocarcinoma according to biomarker categories. Model 1 was adjusted for age and sex and included all participants. Models 2 and 3 were additionally adjusted for performance status (0–1 and 2–4), operation (yes/no), cancer stage (I–II and III–IV) and all the biomarkers (except if considered exposure). Model 2 included patients with complete information and Model 3 all patients due to multiple imputation.

**Figure 3 cancers-13-04599-f003:**
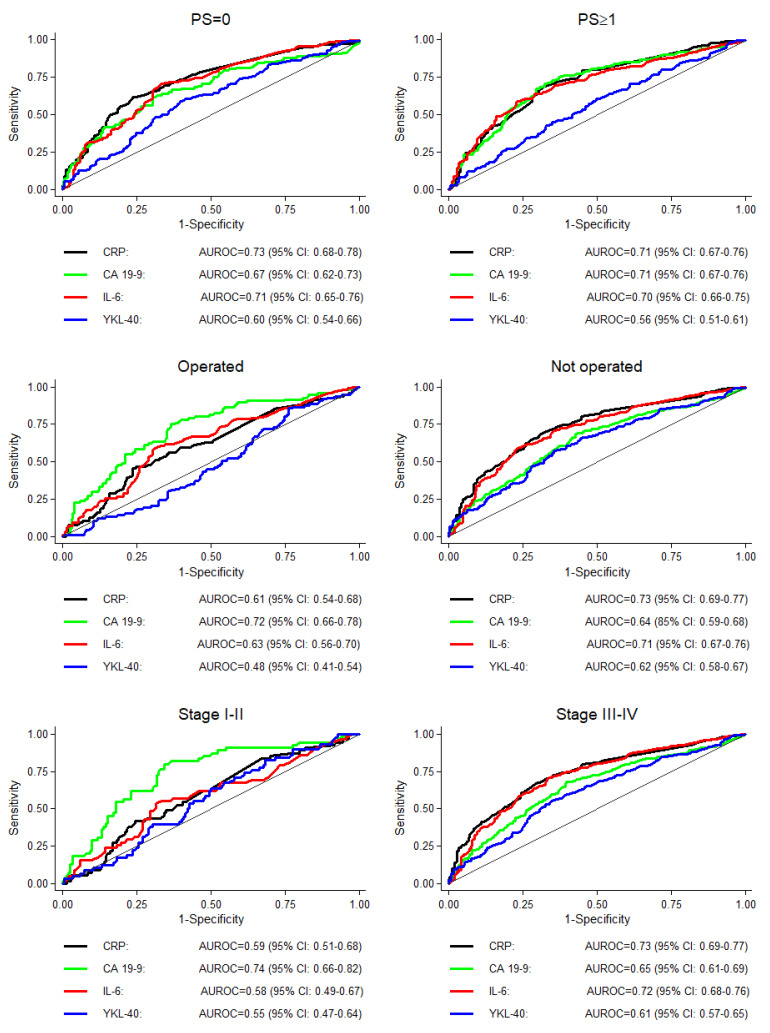
Area under the receiver operating characteristic (AUROC) curves for one-year mortality for CRP, CA 19-9, IL-6 and YKL-40, stratified by performance status (PS), operation (yes/no) and pancreatic ductal adenocarcinoma stage.

**Figure 4 cancers-13-04599-f004:**
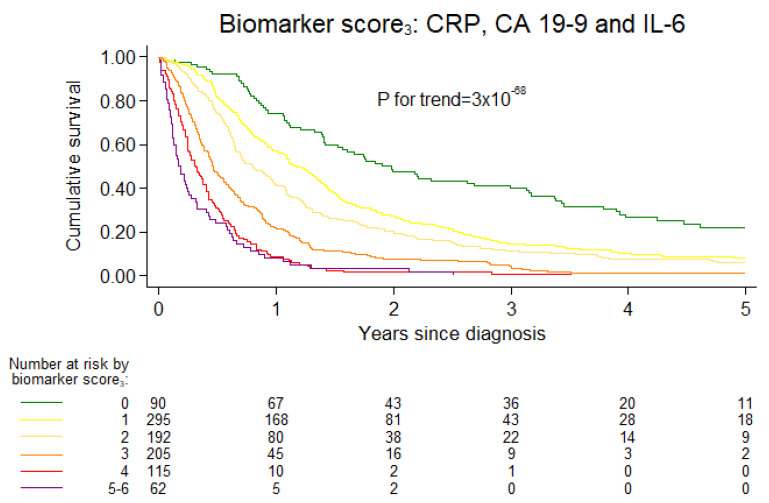
Kaplan–Meier survival curve for plasma CRP, CA 19-9 and IL-6 categories combined into a biomarker score_3_. Biomarker score_3_ was calculated as the sum of CRP, CA 19-9 and IL-6 categories coded as 0, 1 and 2 for low, intermediate and high plasma levels, respectively. *p*-value for trend is from Wald test of trend across groups.

**Figure 5 cancers-13-04599-f005:**
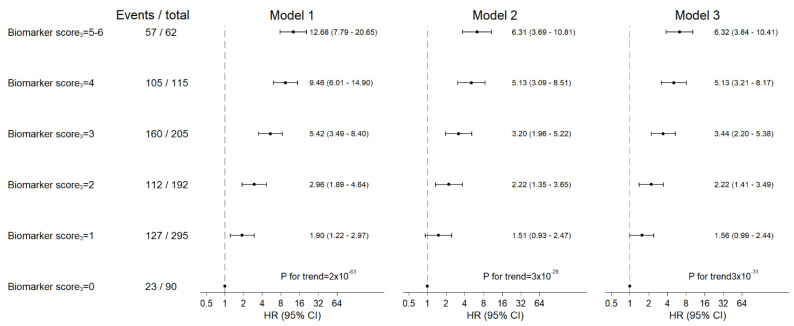
Risk of one-year mortality after pancreatic ductal adenocarcinoma according to biomarker score_3_. Biomarker score_3_ was calculated as the sum of CRP, CA 19-9 and IL-6 categories coded as 0, 1 and 2 for low, intermediate and high plasma levels, respectively. *p*-values for trend are from Wald tests across groups treating biomarker levels as continuous variables in the Cox regression model.

**Table 1 cancers-13-04599-t001:** Characteristics of patients with pancreatic ductal adenocarcinoma according to survival.

Patient Characteristics	All	≤1 Year Survival	>1 Year Survival	Missing, %
Number of participants, N (%)	993 (100%)	603 (61%)	390 (39%)	NA
Age, years	67 (61–72)	67 (61–73)	67 (61–72)	0
Women, %	45	44	48	0
Eversmoker, %	61	61	62	10
Body mass index, kg/m^2^	23.1 (20.7–25.8)	23.2 (20.7–25.9)	22.8 (20.7–25.7)	5.8
High alcohol consumption, %	23	22	23	10
Performance status	1 (0–1)	1 (0–1)	0 (0–1)	9.7
Charlson comorbidity index	3 (2–4)	3 (2–4)	3 (2–4)	3.7
Diabetes, %	26	26	25	1.8
Operated, %	30	16	52	0
Metastasing cancer, %	50	65	27	0.5
Tumour size, mm	3.5 (2.5–4.8)	4.0 (3.0–5.0)	3.0 (2.3–4.0)	13
Stage I–II, %	21	9.6	39	1.5
Stage III, %	27	23	34	0.8
Stage IV, %	51	66	27	0
C-reactive protein, mg/L	9 (3–37)	17 (5–52)	4 (3–11)	1.8
CA 19-9, kU/L	445 (59–3126)	997 (148–6290)	105 (29–796)	1.6
Interleukin-6, µg/L	5.4 (2.7–12)	7.2 (3.9–16)	3.3 (2.0–6.1)	0
YKL-40, µg/L	138 (81–227)	156 (92–246)	113 (67–205)	0

Values collected at inclusion from 3 July 2008 to 24 August 2017, are expressed as numbers of participants, frequencies or medians (interquartile ranges). Numbers of participants across listed variables vary slightly because we did not have information on all participants for all listed variables. NA = not applicable.

## Data Availability

The data presented in this study are not publicly available due to the Danish legislature, e.g., for the privacy of individuals that participated in the study.

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
