# Peer review of "Inflammatory Biomarker Score Identifies Patients with Six-Fold Increased Risk of One-Year Mortality after Pancreatic Cancer"

_cancers, 2021, doi:10.3390/cancers13184599_

Round 1

Reviewer 1 Report

By the US National Comprehensive  Cancer  Network  guidelines, plasma CA 19-9 has remained the most widely and the only biomarker currently recommended for clinical use. Authors' study present a new concept A combination of CRP, CA 19-9 and IL-6 into a single biomarker score , was the best prognostic marker, and may be clinically valuable in identifying patients with PDAC who have the poorest prognosis. In addition, authors demonstrated that CRP out-performed CA 19-9 for prognostic purposes in the 70% of the patients with PDAC who were not operated. It's quite new and important that the authors provided the new evidence that there're new biomarker score for PDAC patients. However, this study is based on 993 adult patients in Denmark, please try to discuss the possibility about the different findings in different countries or racial differences in the Discussion session. 

Author Response

Thank you for the positive evaluation of our manuscript and for your help in the reviewing process.

We have now added the following to the Limitation section (lines 349-352): “Finally, as we only included patients from the (predominantly white) Danish population, our findings do not necessarily apply to other ethnicities. Yet, to our knowledge, there are no data suggesting that our findings could not be directly applicable to other ethnicities as well.”

Reviewer 2 Report

The manuscript is well written and well structured.

The authors propose an inflammatory biomarker score to identify pancreatic cancer patients with an increased risk of one-year mortality. They show that a simple combination of blood tests (CRP, CA 19-9 and IL-6) into a single biomarker score was a better marker of one-year survival than the currently recommended CA 19-9 alone.

The topic is interesting. Future validation studies are needed to confirm their results.

A careful reading is recommended to fix some small typos (e.g. the number of patients in the Introduction at line 58: “in 7-474 patients with pancreatic cancer”).

In my opinion, the article can be published.

Author Response

Reviewer #2

The manuscript is well written and well structured.

The authors propose an inflammatory biomarker score to identify pancreatic cancer patients with an increased risk of one-year mortality. They show that a simple combination of blood tests (CRP, CA 19-9 and IL-6) into a single biomarker score was a better marker of one-year survival than the currently recommended CA 19-9 alone.

The topic is interesting. Future validation studies are needed to confirm their results.

Response: Thank you for the positive evaluation of our manuscript and for your help in the reviewing process. We fully agree that future validation studies are needed to confirm our results, as mentioned under the “Simple Summary” and the Limitations section.

A careful reading is recommended to fix some small typos (e.g. the number of patients in the Introduction at line 58: “in 7-474 patients with pancreatic cancer”).

Response: While 7 may look like a typo, it is in fact not. Table 3 in Vainer’s review (ref 10, also found here: https://pubmed.ncbi.nlm.nih.gov/30038723/) refers to Tsakinaga 2015 study (https://pubmed.ncbi.nlm.nih.gov/26494971/) with 7 patients. One of the co-authors had the same comment, which is understandable as 7 patients is a very small cohort. We have now changed the wording to “in from only seven up to 474 patients with pancreatic cancer”.

Furthermore, as recommended, we read through the manuscript again, and made small changes in the Introduction section mainly regarding acronyms.

In my opinion, the article can be published.

Response: Thank you so much.

Reviewer 3 Report

Although the topic is not very novel the article by Kjaergaard et al. on inflammatory biomarkers in pancreatic cancer is well written, the study design is concise and the rather high number of patients included is a strength.

Questions and Comments:

  • The quality of the figures should be improved as especially the text within the figures is, at least in parts, illegible.
  • Were the patients in the study included prospectively at the time of diagnosis? What was the median time from evaluation of lab values to intiation of treatment? Were there any measures taken to make sure that high CRP or Il-6 values were not caused by infection but were actually a sign for more aggressive/inflammatory tumors?
  • At how many different centres were the patients treated and were there any differences in patients characateristics or in the results of biomarker analyses according to centre?
  • Both, the introduction and the discussion are rather narrow and could be improved.  While CA19-9, CRP and IL-6 are commonly used in clinical routine, YKL-40 is not as known. I would suggest to explain in the introduction. The discussion could be improved by an interpretation of possible biological explanations for the study results. An explanation for the high preditivevalue of Ca 19-9 in operated patients could be that patients actually had metastastic disease that was missed at initial staging. Which staging Tests were done? Was  there a difference in the levels of CA19-9 in operated patients that were staged with PET-CT or conventional CT?

Author Response

Reviewer #3

Although the topic is not very novel the article by Kjaergaard et al. on inflammatory biomarkers in pancreatic cancer is well written, the study design is concise and the rather high number of patients included is a strength.

Questions and Comments:

  • The quality of the figures should be improved as especially the text within the figures is, at least in parts, illegible.

Response: We have revised the figures as suggested.

  • Were the patients in the study included prospectively at the time of diagnosis? What was the median time from evaluation of lab values to initiation of treatment? Were there any measures taken to make sure that high CRP or Il-6 values were not caused by infection but were actually a sign for more aggressive/inflammatory tumors?

Response:

Yes, the patients were included prospectively at the time of diagnosis. We have added the following to the “2. Materials and Methods” section: “The BIOPAC study is a prospective multicenter open cohort study with ongoing enrollment.”

Blood was drawn just prior to the commencement of chemotherapy (median [interquartile range]: 0 [0-0] days) or operation (1 [1-34] days). In order to clarify this, we have revised 2.1. Biochemical analyses section as follows: “Blood for biochemical analyses was drawn around the time of diagnosis. For patients with stage II-IV PDAC, blood was drawn just prior to the commencement of chemotherapy (median [25th-75th percentile]: 0 [0-0] days). For patients who were operated (≈30%), it was drawn prior to the operation (0 [0-34] days).”

No measures were taken to ensure that high inflammatory biomarkers were a sign of infection rather than an aggressive PDAC. To address this concern, we have added following to the Limitation section (lines 345-349): “Furthermore, not only PDAC, but also many infections are associated with increased plasma levels of inflammatory biomarkers. However, since blood sampling was performed just prior to the commencement of chemotherapy or operation, both of which are contra-indicated in patients with overt infections, it is unlikely that infections affected plasma biomarker levels to a large degree.”

  • At how many different centers were the patients treated and were there any differences in patients’ characteristics or in the results of biomarker analyses according to center?

Response: We have added the following paragraph to the “2. Materials and Methods” section (lines 81-91): ”The BIOPAC study is a prospective multicenter open cohort study with ongoing enrollment. The patients in this study were enrolled at five hospitals: Copenhagen University Hospital at Herlev (50%, houses the biobank), Hillerød Hospital (6%), Zealand University Hospital at Næstved (9%), Odense University Hospital (10%) and Copenhagen University Hospital at Rigshospitalet (25%). Because Rigshospitalet is the most highly specialized hospital in Denmark and the largest one in volume with expertise in pancreatic cancer surgery, 71% of patients enrolled at Rigshospitalet were operated. As operated patients usually have smaller tumor burden and better prognosis than patients who are not operated, this likely explains why patients enrolled at Rigshospitalet had somewhat lower plasma levels of inflammatory biomarkers. However, no other patient characteristics differed between the hospitals.”

Please note that although the above can be considered discussion of results, we purposely included it in the Materials and Methods section because it describes the cohort and not the main results, and because it is easier for the reader to follow when explained in one coherent than several different sections.

  • Both, the introduction and the discussion are rather narrow and could be improved.  While CA19-9, CRP and IL-6 are commonly used in clinical routine, YKL-40 is not as known. I would suggest to explain in the introduction. The discussion could be improved by an interpretation of possible biological explanations for the study results. An explanation for the high predictive value of CA 19-9 in operated patients could be that patients actually had metastatic disease that was missed at initial staging. Which staging Tests were done? Was there a difference in the levels of CA19-9 in operated patients that were staged with PET-CT or conventional CT?

Response:

While we agree that our Introduction and Discussion sections are brief (which was intentional on our part), we believe that they also are concise. This is, of course, a matter of personal preference.

We also agree that YKL-40 is less known than the other biomarkers. We have now added “(See Supplementary Introduction)” after the first mention of YKL-40 in the Introduction section. As YKL-40 is not included in the final biomarker score, we believe that it is more appropriate to present details as Supplementary Material than in the main text.

We agree that very high CA 19-9 levels may indicate undetected advanced PDAC. We have now added the following to the Limitation section (lines 340-345): “According to the Danish national guidelines, conventional CT scan is the most commonly used modality at initial staging of PDAC. Thus, microscopic disease and small metastases under detection level may have been overlooked. Since very few patients had additional PET-CT performed, we were unable to investigate CA 19-9 levels according to CT modality (conventional versus PET-CT).”

However, the presence of metastatic disease cannot explain the high predictive value of CA 19-9 in operated patients. In fact, CRP in operated patients may be elevated due to other causes than PDAC, and may therefore be slightly inferior to CA 19-9, which is more sensitive to PDAC than infection. We have now added following to the Discussion section (lines 315-318): “A possible explanation could be that CRP and IL-6 compared with CA 19-9 may be more sensitive biomarkers of other inflammation than PDAC, e.g., infections. However, this is only speculative.”

Furthermore, we added two more limitations to the Discussion section according to reviewer comments (lines 345-352): “Furthermore, not only PDAC, but also many infections are associated with increased plasma levels of inflammatory biomarkers. However, since blood sampling was performed just prior to operation or commencement of chemotherapy, both of which are contra-indicated in patients with overt infections, it is unlikely that infections affected plasma biomarker levels to a large degree. Finally, as we only included patients from the (predominantly white) Danish population, our findings do not necessarily apply to other ethnicities. Yet, to our knowledge, there are no data suggesting that our findings could not be directly applicable to other ethnicities as well.”